# OpenReview forum: "Erased, But Not Forgotten: Erased Rectified Flow Transformers Still Remain Unsafe Under Concept Attack"
_ICLR.cc/2026/Conference — ICLR 2026 Conference Withdrawn Submission_

### Official Review · Reviewer_KpQF · 2025-10-29

**Soundness:** 3
**Presentation:** 3
**Contribution:** 2
**Rating:** 4
**Confidence:** 3

**Summary:**

This paper proposes a concept restoration attack (ReFlux) against concept erasure on the FLUX model, while prior research primarily focused on U-Net diffusion models. This paper motivates its method by pointing out that previous approaches are not directly applicable to the FLUX model, as they were designed for earlier architectures. In contrast to existing methods like UnlearnDiffAtk, ReFlux fine-tunes the model again, which makes the comparison to prior works (such as discrete black-box attacks) less clear. It has already been demonstrated many times that any downstream fine-tuning makes erasure effects slowly disappear; however, it remains unclear whether such benign fine-tuning would have similar effects to the proposed dedicated ReFlux fine-tuning. Unfortunately, the performance difference between UnlearnDiffAtk and the original model is usually fairly small, considering that UnlearnDiffAtk does not modify the model weights again. The inversion-based CCE attack is missing as a baseline. However, the general execution of this study is very well done, while the writing is a bit over-formalised. Novel contributions on the attack side appear to be limited, except for making different assumptions than prior works and specifically focusing on FLUX, while related work studied earlier diffusion models. Nevertheless, it is the first study on adversarial robustness of concept erasure methods on the FLUX model.

**Strengths:**

- (S1) **First work on concept restoration on FLUX** and generally more modern rectified flow models.
- (S2) **Many illustrative samples and illustrations in the main paper** such as Figures 2, 3, and 6.
- (S3) **Detailed derivation of why existing attacks struggle on newer architectures**, which then serves as a motivation for the proposed ReFlux approach.

**Weaknesses:**

I find the following list of things to be major weaknesses:
- (W1) **A crucial new assumption** that the attacker can fine-tune the model, which is not the case in existing works on concept restoration attacks (like CCE). The usual setting is that only the inputs to the model are perturbed or optimised in a way such that the model regenerates previously erased content. This new assumption significantly limits the practical implications of this work.
- (W2) **Is this method better than just fine-tuning on any data downstream?** - Many works already showed that erasure methods lack robustness against benign fine-tuning on any data, mirroring the catastrophic forgetting from continual learning as a catastrophic forgetting-to-suppress in the concept erasure area. The key question is whether such a normal fine-tuning, which shares the exact same assumptions as ReFlux, performs similarly and how ReFlux compares to it. Of course, ReFlux is more efficient, but it also assumes that the attacker can access the model after unlearning.
- (W3) **Difference in the performance to UnlearnDiffAtk is usually small**: The ReFlux attack is not always outperforming UnlearnDiffAtk, despite making the crucial assumption of being able to fine-tune the model.

The following one is a more minor weakness:
- (W4) **Important missing attack baseline**: *"Circumventing Concept Erasure Methods For Text-to-Image Generative Models (CCE)"* by Pham et al. (2023) is missing as an important, inversion-based attack. It could be applied only to the CLIP encoder, for example.

**Questions:**

- (Q1) **No highlighting** of best results in Tables 2 and 3?
- (Q2) **Unclear Attack-guided velocity loss**: Is the intuition basically just applying the ESD negative guidance distillation again against an (ESD) erasure by reverting the negative guidance back to positive guidance in a sense?
- (Q3) **Why is the LoRA Consistency Loss called "LoRA"?** The loss could also be applied without PEFT/LoRA, right?

---

### Official Review · Reviewer_hzY1 · 2025-10-30

**Soundness:** 2
**Presentation:** 3
**Contribution:** 2
**Rating:** 4
**Confidence:** 4

**Summary:**

This paper proposes ReFlux, the first concept attack method specifically designed for rectified flow–based text-to-image (T2I) models, such as Flux. The authors observe that existing concept erasure methods in Flux rely on an attention localization mechanism. Based on this, they develop a lightweight attack approach that reactivates erased concepts by fine-tuning a small set of text-related parameters via LoRA. The method combines three objectives: (1) an attention reactivation loss for restoring suppressed tokens, (2) a velocity-guided loss to steer generation dynamics, and (3) a consistency loss to preserve unrelated content. Experiments across various concept types (e.g., nudity, violence, style, abstraction) show that ReFlux outperforms baseline attacks, revealing that erased concepts can still be recovered in Flux.

**Strengths:**

1. The authors clearly identify attention localization as a key property in Flux’s erasure mechanism and design an attack that directly targets it, showing strong conceptual motivation.

2. Experiments are comprehensive, covering multiple types of concepts (NSFW, style, relationship, etc.) and showing strong performance across all cases.

**Weaknesses:**

1. My **major concern** is the task setting: ReFlux modifies the model’s own parameters via LoRA. This makes it closer to retraining with concept supervision than a traditional attack. It assumes full weight access and update permission, which is not realistic in many deployment settings.

2. The paper critiques prior black-box or LLM-based attacks for being "unstable" or "slow", but those methods operate under much stricter access assumptions (e.g., API-only, or even under white-box settings, they do not change the model parameter), which are more representative of real-world scenarios. In contrast, ReFlux assumes white-box access and permission to train the model, which is a much stronger and less realistic assumption.

3. From my understanding, it is obtained by extracting the tokens in the prompt that are related to the target concept. However, in erased models, does the attention between the text tokens and the visual tokens corresponding to the target concept still remain? In some anchor-based methods, new attention may be constructed between newly introduced text tokens and the visual tokens of the target concept.

4. Missing related works:

[1] Concept corrector: Erase concepts on the fly for text-to-image diffusion models

[2] Dark miner: Defend against unsafe generation for text-to-image diffusion models

[3] One Image is Worth a Thousand Words: A Usability Preservable Text-Image Collaborative Erasing Framework

[4] Erasing More Than Intended? How Concept Erasure Degrades the Generation of Non-Target Concepts

**Questions:**

1. The method assumes full white-box access and the ability to fine-tune the model’s internal parameters using LoRA. In what real-world deployment scenario would such access be available to an attacker? How does this setting align with actual safety concerns in deployed T2I systems?

2. Since ReFlux modifies internal weights using supervised loss objectives, the process is conceptually similar to adversarial fine-tuning. What safeguards or constraints distinguish this method from just re-training the model with new prompts that contain the target concept?

3. I believe the observed attention localization is a universal feature in erased models. But can merely focusing on this reactivate the erased concept? What if the erasing method optimizes the UNet / Stream Block (not only cross-attention-like layers) or the text encoder? In these cases, the model may totally lose the ability to understand the target concepts (updating text encoder) or totally lose the ability to construct visual content of the target concepts (updating Stream Block, not only cross-attention-like layers).

4. Can this method be applied back to SD?

If the authors can solve the questions, I am willing to raise my rating.

---

### Official Review · Reviewer_GqVL · 2025-10-31

**Soundness:** 3
**Presentation:** 3
**Contribution:** 3
**Rating:** 6
**Confidence:** 4

**Summary:**

This paper investigates how to circumvent concept erasure in modern rectified flow–based text-to-image models (e.g., Flux). The authors argue that existing erasure and attack techniques designed for Stable Diffusion transfer poorly to Flux, largely because of architectural differences, especially the lack of explicit cross-attention layers. They observe that concept erasure on Flux implicitly relies on attention localization, and based on this insight propose ReFlux, a lightweight LoRA-based fine-tuning attack that reactivates the erased concept. Experiments demonstrate that ReFlux can consistently recover suppressed concepts, making it a useful benchmark for assessing the robustness of concept-erasure strategies on rectified flow models.

**Strengths:**

- **Clear motivation**.This work is self-motivated. It explains why SD-based erasure pipelines transfer poorly to rectified-flow transformer T2I models (e.g., Flux) and underscores the absence of a reliable robustness benchmark in this setting.
- **Mechanistic insight**. It identifies attention localization as the core mechanism behind concept erasure in Flux-like models, offering meaningful, practical guidance for future algorithm design (both defenses and attacks).
- **Promising evidence with lightweight tuning**. Empirical results convincingly support the effectiveness of ReFlux, implemented as a lightweight LoRA text-side fine-tune (~3.57 MB).

**Weaknesses:**

1. **Limited generalization beyond Flux**. This paper is concentrated on Flux.1, eaving it unclear how well the findings transfer to other rectified-flow T2I variants (different backbones/scales, text encoders, or training recipes).
2. **A little strong access assumptions**.The attack assumes white-box access sufficient for LoRA fine-tuning of text-side parameters and for reading attention signals, which may hinder practical use in API-only or weight-restricted deployments.

3. **Insufficient efficiency comparison**. Although the paper highlights an inference-time advantage for ReFlux, the method still requires a dedicated LoRA fine-tuning phase. The efficiency analysis largely benchmarks only the inference stage, omitting end-to-end training overhead.

**Questions:**

Overall, this is an interesting submission with clear motivation, a lightweight implementation, and convincing empirical evidence. The questions below are intended to strengthen its practical impact, and the authors are encouraged to address the major concerns first.
- Major:

1. How sensitive is performance to key hyperparameters—especially the various weight $\lambda$ in Eq. (6)?

2. Is the reactivation effect stable if fine-tuning continues longer (or across multiple sessions)?
- Minor:

1. Does ReFlux transfer to other rectified-flow T2I models beyond Flux.1?
2. What defense strategies do the authors recommend, and can randomized or multi-layer erasure (e.g., stochastic masking) mitigate ReFlux?

---

### Official Review · Reviewer_AwtA · 2025-11-06

**Soundness:** 3
**Presentation:** 3
**Contribution:** 2
**Rating:** 2
**Confidence:** 4

**Summary:**

This paper investigates the robustness of concept-erasure defenses in next-generation rectified-flow transformer–based text-to-image models (notably Flux). The authors propose ReFlux, the first dedicated concept-attack method for this architecture.
The key insight is that existing erasure techniques, when ported from Stable Diffusion (SD) to Flux, ultimately depend on attention localization, i.e., suppression of token-indexed attention maps. Building on this, ReFlux introduces a reverse-attention optimization mechanism stabilized through (i) an attention-reactivation loss with L2 and entropy regularization, (ii) a velocity-guided objective that steers the rectified-flow trajectory to reinforce erased concepts, and (iii) a LoRA-based consistency constraint that preserves unrelated content and layout.
Extensive experiments on NSFW, artistic style, entity, abstraction, and relational benchmarks show that ReFlux achieves strong reactivation of erased concepts while maintaining global fidelity, outperforming prior prompt-based and gradient-based attack baselines.

**Strengths:**

1. The paper tackles a underexplored topic: evaluating safety and erasure robustness in rectified flow transformers (Flux).
2. The idea of analyzing attention localization as the root of erasure behavior is interesting.
3. The paper is well written.
4.The Experiments cover several datasets and concept categories (I2P, ConceptPrune, CelebA, artistic style).

**Weaknesses:**

1. Lack of novelty in the core idea:

Despite the framing, the proposed “reverse-attention optimization” is essentially a restatement of gradient-based attention amplification with regularization terms (L2 and entropy). The formulation in Eq. (6) combines standard penalties already seen in prior works like ESD. There is no new learning principle introduced.

2. No formal analysis of “attention localization:

The paper repeatedly claims that all erasure methods on Flux rely on “attention localization,” but provides only qualitative heatmaps. No quantitative correlation analysis, ablation on token indices, or mathematical evidence is offered to prove this is a necessary mechanism. The argument remains largely descriptive.

3. The link between velocity guidance and flow-matching dynamics is hand-wavy:

Equation (7) treats the velocity loss as an analog to v-prediction (Salimans & Ho, 2022), but the derivation is vague and lacks any justification for why this perturbation reliably reactivates erased semantics. The loss looks more like a heuristic energy term than a well-motivated generative control mechanism.

4. No clear understanding of how LoRA affects erasure behavior:

The use of LoRA fine-tuning on text parameters is presented as “consistency preservation,” but the paper gives no empirical or theoretical evidence that this term actually prevents drift or stabilizes semantics. It is more likely that the LoRA term simply reduces attack strength due to smaller parameter capacity.

5. Weak evaluation design:

The evaluation mostly uses automatic classifiers (NudeNet, Q16, CLIP, ViT), any human verification is missing. The authors claim “high-fidelity recovery” but present no quantitative perceptual metrics (FID, CLIP similarity).

6. No comparison with stronger or more recent safety methods:

The erasure baselines stop at ESD, AC, and CP. There is no comparison to more advanced safety-tuned rectified-flow checkpoints or concept-regulated fine-tuning methods (e.g., MACE or SAFEDiff). The evaluation therefore does not convincingly show that ReFlux is effective against state-of-the-art defenses.

7. Limited scope of the model base:

All experiments use Flux.1 [dev], a single open-source checkpoint. It is unclear whether the proposed attack generalizes to Flux [schnell], SD1.4/2, or other flow-based architectures.


8. Efficiency claims are overstated:

The authors state that ReFlux requires “only one image generation (~0.5 min)” but fine-tunes LoRA for 1000 steps. This is contradictory — the cost of optimization per concept is not negligible. Compared to Ring-A-Bell (which is training-free), ReFlux is actually more computationally expensive per concept.


9. Clarity and Presentation Issues:

The writing, while technical, is dense and overly formal, with many equations that add little insight (e.g., rewriting entropy and KL expansions just to derive Eq. (6)). Some notation is inconsistent (switching between 𝑣𝜃, 𝑣, 𝑢𝑡) and several equations lack context for variables like 𝑥𝑇,𝑢pix​.
I feel the figures use “blue bars” and “yellow frames” are covering less area the most parts are bit revealing.

**Questions:**

1. How does the method perform when the erased concept involves multi-word expressions or abstract attributes (e.g., “freedom” or “violence in abstract art”)?

2. Did the authors test the attack on multiple seeds or random initializations to verify stability?

3. Could the improvements simply arise from fine-tuning overfitting rather than a genuinely targeted reactivation?

---

### Note · Authors · 2025-11-13

**Comment:**

Thanks to all reviewers and Area Chairs for your valuable comments and insights. Regarding the initial scores, we have decided not to forgo the rebuttal phase.

Best regards.

**Withdrawal Confirmation:**

I have read and agree with the venue's withdrawal policy on behalf of myself and my co-authors.